# TOWARDS OFF-ROAD AUTONOMOUS DRIVING VIA PLANNER GUIDED POLICY OPTIMIZATION

## ABSTRACT

Off-road autonomous driving poses significant challenges such as navigating diverse terrains, avoiding obstacles, and maneuvering through ditches. Addressing these challenges requires effective planning and adaptability, making it a long-horizon planning and control problem. Traditional model-based control techniques like Model Predictive Path Integral (MPPI) require dense sampling and accurate modeling of the vehicle-terrain interaction, both of which are computationally expensive, making effective long-horizon planning in real-time intractable. Reinforcement learning (RL) methods operate without this limitation and are computationally cheaper at deployment. However, exploration in obstacle-dense and challenging terrains is difficult, and typical RL techniques struggle to navigate in these terrains. To alleviate the limitations of MPPI, we propose a hierarchical autonomy pipeline with a low-frequency high-level MPPI planner and a high-frequency low-level RL controller. To tackle RL's exploration challenge, we propose a teacher-student paradigm to learn an end-to-end RL policy, capable of real-time execution and traversal through challenging terrains. The teacher policy is trained using dense planning information from an MPPI planner while the student policy learns to navigate using visual inputs and sparse planning information. In this framework, we introduce a new policy gradient formulation that extends Proximal Policy Optimization (PPO), leveraging off-policy trajectories for teacher guidance and on-policy trajectories for student exploration. We demonstrate our performance in a realistic off-road simulator against various RL and imitation learning methods. Source code and videos are available at this link.

## 1 INTRODUCTION

Autonomous ground vehicles have advanced significantly in recent years, with applications such as delivery robots and self-driving taxis. While great progress has been made in structured, urban environments, navigating off-road terrains remains a major challenge. Unlike on-road driving, off-road driving requires effective planning to avoid obstacles, speed management to navigate extreme slopes, and rapid adaptive maneuvers to handle varied traction levels and terrains such as dirt, sand, and rocks. Hence, it requires sophisticated control techniques to traverse in these challenging terrains. Successfully navigating large, unstructured environments also depends on effective long-distance planning, making it both a long-horizon planning and adaptive control problem.

Conventional control methods for off-road vehicles often depend on model-based techniques, like Model Predictive Path Integral (MPPI) (Han et al. (2024); Williams et al. (2015)). Model-based techniques rely on environmental details, such as segmentation maps and depth maps, to provide waypoints for a low-level controller. These methods necessitate very dense sampling of waypoint rollouts to effectively avoid obstacles and manage diverse terrains. However, this dense sampling requirement is computationally expensive, making it impractical to run these controllers in real-time for simultaneous globally optimal trajectory planning and terrain handling. Some attempts have been made to improve sampling efficiency by learning a state-dependent control action distribution and learning a terminal value function (Qu et al. (2024); Hansen et al. (2022)), thereby reducing required number of samples and planning horizon.

Reinforcement learning (RL) is highly effective for tackling complex, high-dimensional, and sequential tasks that are often challenging for traditional control methods. RL models typically utilize

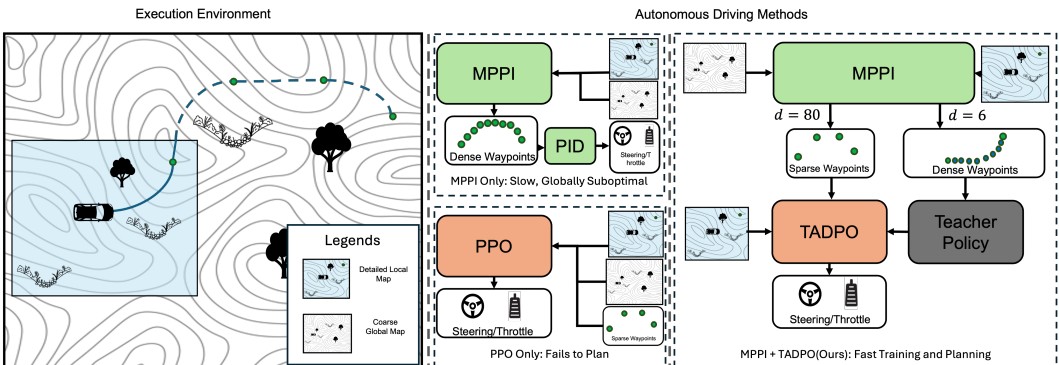

Figure 1: Illustration of the proposed hierarchical autonomy framework, integrating an MPPI planner and an RL controller for off-road navigation. During deployment, the framework enables effective global planning through MPPI while reducing the need for frequent costly sampling. During training, the planner plans at different granularity, facilitating training of a teacher policy using dense waypoints. The teacher's demonstrations facilitate effective exploration during the training of the student policy through updates provided by TADPO.

neural networks with a limited number of layers, enabling rapid inference. This capability also allows them to execute swift action maneuvers effectively in response to diverse terrains which is impossible with model-based planners. However, in an off-road environment, when attempting to avoid obstacles and ditches, RL methods face significant challenges in exploration, often rendering it difficult to learn these complex tasks effectively. Specifically in applications like off-road autonomous driving where environment simulation is relatively costly, environment transition dynamics are highly stochastic, and dense rewards can encourage expediency, exploration is hard without guidance from an external source or access to global planning information. Some works have explored end-to-end RL methods (Kalaria et al. (2024); Hensley & Marshall (2022); Wang et al. (2023)) for specific aspects of off-road driving. However, they lack a planning component and realistic simulation, making these methods significantly less suitable for realistic off-road autonomy. Some works (Kendall et al. (2018); Isele et al. (2018); Fayjie et al. (2018)) have attempted to address specific aspects of on-road driving using RL. However, the challenge there is the unpredictable behavior of the other actors rather than the variability of the terrain.

Proximal Policy Optimization (PPO) proposed by Schulman et al. (2017) is a popular RL framework that allows for stable on-policy learning. Despite its advantages, PPO faces limitations in effective exploration as it relies on random actions sampled around the policy's intended action for exploration. Because of this, in the proposed off-road driving problem, training a policy with PPO for waypoint distances greater than tens of meters faces significant exploration challenges, and attempting to master multiple off-road navigation skills particularly in avoiding obstacles and navigating extreme slopes leads to ineffective policy training and failure to complete the task. Therefore, our goal is to distill planning information from a teacher trained with the aid of a dense planner on a reduced observation space, while a student learns off-road traversal using an extended observation space without access to the computationally expensive planning data.

Since PPO is on-policy, it can only be trained on trajectories collected from its own policy and cannot incorporate external guidance. Attempts to incorporate demonstrations in PPO have been made, though with notable limitations. PPO+D (Libardi & Fabritiis (2021)) extends PPO by incorporating a single off-policy trajectory into the training process. This approach modifies the PPO replay buffer to include three components: $D_r$ for successful trajectories, $D_v$ for failure trajectories and $D$ for the currently sampled trajectories. When sampling from $D_v$, the paper employs value-based sampling, which becomes impractical for tasks that involve large replay buffers with visual inputs. In off-road driving, navigating diverse terrains requires a broad range of skills, making large buffers for the teacher demonstration replay and the failure replay buffer in PPO+D crucial. This necessity renders PPO+D unsuitable for the task.

There have been a few attempts to learn policies from demonstrations and through a teacher-student framework in autonomous vehicles. Peng et al. (2022) use off-policy methods like SAC and choose actions between a teacher and student policy to solve simpler tasks like lane following and obsta-

cle avoidance. Some teacher-student paradigms, such as those using Deep Q-network Hester et al. (2017), focus on discrete action spaces. Kang et al. (2018); Martin et al. (2021) use existing off-policy methods like Soft Actor-Critic (SAC) to update a student policy. In complex planning and control tasks, these methods tend to be less stable during training (James W. Mock (2023)). Therefore, we use PPO as our RL training method.

To address the limitations of MPPI, we propose a hierarchical end-to-end pipeline that integrates a high-level MPPI planner with a lower-level reinforcement learning controller focused on adaptive execution for effective obstacle avoidance and navigation through challenging terrains. To resolve the exploration issues of PPO, we propose a novel method, Teacher Action Distillation with Policy Optimization (**TADPO**), which extends PPO to optimize policy based on trajectories collected by an expert teacher policy.

## 2 BACKGROUND

We formulate the low-level control of the off-road driving problem as a Markov Decision Process (MDP), represented by the tuple $\mathcal{M} = (\mathcal{S}, \mathcal{A}, P, r, \gamma)$, where: $\mathcal{S}$ is the state space, $\mathcal{A}$ is the action space, $P(s'|s, a)$ is the transition dynamics function, $r : \mathcal{S} \times \mathcal{A} \to \mathbb{R}$ is the reward function, and $\gamma \in [0, 1)$ is the discount factor. Our objective is to identify an optimal policy $\pi^*$ such that

$$\pi^* = \arg \max_\pi \mathbb{E}_\pi \left[ \sum_{t=0}^\infty \gamma^t r(s_t, a_t) \right] \tag{1}$$

### 2.1 POLICY GRADIENT OPTIMIZATION METHODS

A common family of on-policy RL methods uses a policy gradient to optimize policies. A key aspect of policy gradient methods, is that the gradient is computed with respect to the distribution of states induced by the current policy. By utilizing this distribution, policy gradients can be derived from the expected return, facilitating updates to the policy parameters. In general, the policy gradient has the form:

$$\nabla J(\theta) = \mathbb{E}_{\tau \sim \pi_\theta} \left[ \nabla \log \pi_\theta(a_t|s_t) \hat{A}(s_t, a_t) \right] \tag{2}$$

where $\tau$ is a trajectory and $\hat{A}$ is the advantage estimate.

**Proximal Policy Optimization**    PPO, proposed by Schulman et al. (2017), improves traditional policy gradient methods by limiting large policy updates through a clipped surrogate objective to optimize $\theta$:

$$L^{\text{CLIP}}(\theta) = \mathbb{E}_t \left[ \min \left( r_t(\theta) \hat{A}_t, \text{clip}(r_t(\theta), 1 - \epsilon, 1 + \epsilon) \hat{A}_t \right) \right] \tag{3}$$

$$L^{\text{VF}}(\theta) = \mathbb{E}_t \left[ (V_{\pi_{\theta_{old}}}(s_t) - R_t)^2 \right] \tag{4}$$

$$L^{\text{entropy}}(\theta) = \mathbb{E}_t \left[ -H[\pi_\theta(\cdot|s_t)] \right] \tag{5}$$

$$L^{\text{PPO}}(\theta) = L^{\text{CLIP}}(\theta) - c_1 L^{\text{VF}}(\theta) + c_2 L^{\text{entropy}}(\theta) \tag{6}$$

where $r_t(\theta) = \frac{\pi_\theta(a_t|s_t)}{\pi_{\theta_{old}}(a_t|s_t)}$ is the probability ratio of the action in distribution $\pi_\theta(a_t|s_t)$ and $\pi_{\theta_{old}}(a_t|s_t)$, $\hat{A}_t = \sum_{i=t}^{t+T} (\gamma\lambda)^{i-t} \delta_i$ is the generalized advantage estimate with $\delta_t = R_t + \gamma V_{\pi_{\theta_{old}}}(s_{t+1}) - V_{\pi_{\theta_{old}}}(s_t)$, $R_t = \sum_{i=t}^{t+T} \gamma^{i-t} r(s_i, a_i) + \gamma^{T-t+1} V(s_{T+1})$ is the discounted return, and $T$ is the number of transitions, $H[\pi_\theta(\cdot|s_t)]$ is the entropy of the policy's action distribution given state $s_t$, and the value function $V_{\pi_{\theta_{old}}}(s_t)$ is the expected return of state $s_t$. $L^{\text{PPO}}$ updates the actor towards the more advantageous actions at state $s_t$, and $L^{\text{VF}}$ updates the value function so it represents the expected return of the policy for the current state, $L^{\text{entropy}}$ encourages exploration by the policy and $c_1, c_2$ are constants. Instead of making unrestricted updates to the policy, PPO introduces a clipping mechanism to ensure that policy updates remain within a constrained region, which stabilizes training and leads to more reliable convergence.

It is important to note that the advantage estimate $\hat{A}_t$ reflects how much advantageous the current action $a_t$ is compared to the expected value of the state, represented by $V_{\pi_{\theta_{old}}}$. Thus, the gradient

update from equation 6 is meaningful only when $V_{\pi_{\theta_{\text{old}}}}$ sufficiently represents the expected return of the actor of policy $\pi_{\theta_{\text{old}}}$. This intuition is a crucial insight for equation 10 in our proposed method.

For tasks that encounter exploration challenges due to complex planning requirements, PPO fails to learn effective policies (Libardi & Fabritiis (2021)). Introducing undirected randomness to the actor can lead to inefficient exploration because the random actions may not be strategically aligned with the task objectives. This lack of direction leads the agent to explore suboptimal areas, hindering policy improvement in complex environments where targeted exploration is essential. Therefore, it is necessary to distill this planning knowledge while training PPO using a teacher expert. As PPO is an on-policy algorithm, it lacks the ability to learn from off-policy trajectories in the form of demonstrations. This limits its application in tasks where exploration is difficult.

## 2.2 MODEL-BASED CONTROL

Model Predictive Control (MPC) is a traditional control framework that uses sampling or optimization techniques to minimize a cost function, making it effective for generating control action in complex, nonlinear systems. The optimal action sequence $a^*$ is chosen via

$$a^* = \arg\min_a \sum_{i=0}^{h} C(s_i, a_i) \tag{7}$$

where $C$ is the cost function, $s_i$ is the state, $a_i$ is the action at step $i$ and $h$ is the horizon. Some techniques for selecting optimal actions are Cross-Entropy Method (CEM) Kobilarov (2012) and Model Predictive Path Integral (MPPI) Williams et al. (2015). MPPI is a sampling-based method that applies importance-weighted optimization to generate control outputs. CEM is a sampling-based, iterative optimization technique that refines a probability distribution over control parameters for robust control outputs. MPPI has proven popular in recent literature due to its high parallelizability and speed.

## 3 TADPO: TEACHER ACTION DISTILLATION WITH POLICY OPTIMIZATION

As illustrated in Figure 1, we formulate a new method to train a student policy $\pi$ capable of local execution to be used in conjunction with a sparse global MPPI planner by combining on-policy exploration with off-policy distillation. The same MPPI planner also generates dense waypoints spanning the sparse waypoints to train a teacher policy $\mu$. Demonstrations generated by the teacher policy then provides guidance to facilitate exploration and learning of the student policy.

### 3.1 TEACHER ACTION DISTILLATION POLICY GRADIENT

For a pre-trained teacher policy $\mu$, we define the loss $L$ used for training a student policy $\pi$. This loss is applied exclusively to trajectories sampled from the teacher, meaning that actions are drawn from $\mu$ at each time step, $a_t \sim \mu$.

$$L^{\text{TAD}}(\theta) = L^{\mu}(\theta) + c_2 L^{\text{entropy}}(\theta) \tag{8}$$

$$\rho_t(\theta) = \frac{\pi_\theta(a_t|s_t^\pi)}{\mu(a_t|s_t^\mu)} \tag{9}$$

$$\hat{\Delta}_t = R(a_t, s_t) - V_{\pi_{\theta_{\text{old}}}}(s_t^\pi) \tag{10}$$

$$L^{\mu}(\theta) = \mathbb{E}_{a_t \sim \mu}\left[\max\left(0, \min(\rho_t(\theta), 1 + \epsilon_\mu)\hat{\Delta}_t\right)\right] \tag{11}$$

where $L^{\text{entropy}}$ is defined as in equation 5. Note that the teacher policy $\mu$ and the student policy $\pi$ have distinct observation spaces given the same environment state, denoted by $s_t^\mu, s_t^\pi, s_t$ respectively.

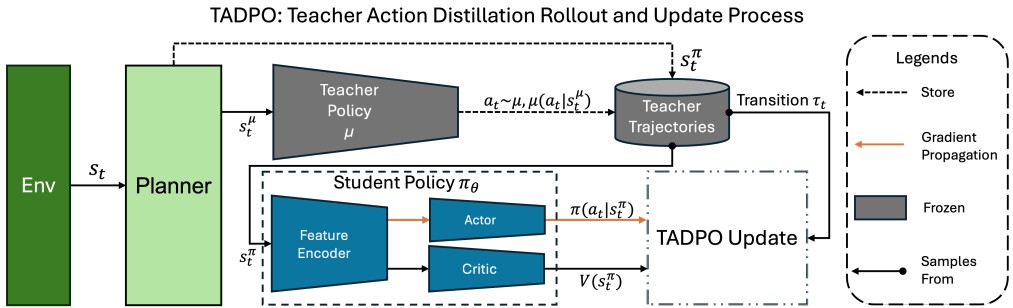

Figure 2: This diagram shows the policy update process when sampling from the teacher demonstration replay buffer. This update process only updates the actor and the feature encoder of the policy and uses the critic as the measure of the relative advantage between teacher action to the student performance given the observation.

The likelihood ratio defined in equation 9 resembles the one used in PPO when $\hat{A}_t > 0$. We substitute the likelihood of $a_t$ under $\pi_{\theta_{\text{old}}}$ with the likelihood under $\mu$. $\rho$ quantifies the difference between $\pi_\theta$ and $\mu$ and clipping $\rho$ restricts gradient updates to the policy when $\rho$ exceeds $1 + \epsilon_\mu$, where $\epsilon_\mu$ is a hyperparameter.

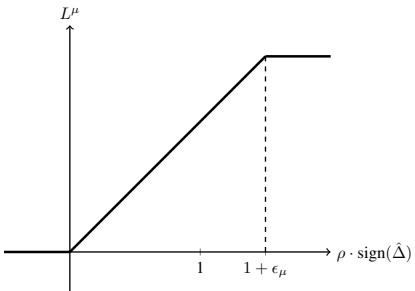

Figure 3: A single timestep of the teacher distillation loss function $L^\mu$ as a function of $\rho * \text{sign}(\hat{\Delta})$. The intended effect of the $L^\mu$ formulation is that student policy only learns from the teacher policy when the return by the teacher demonstration is higher than the expected return of the student given the state and not too much more likely (controlled by clipping factor $\epsilon_\mu$) to predict such action, thereby ensuring stability of policy during training.

In equation 10, $\hat{\Delta}_t$ measures the advantage between the discounted reward from $s_t$ collected using the teacher policy and the expected student policy return $V_{\pi_{\theta_{\text{old}}}}(s_t^\pi)$. As $V_{\pi_{\theta_{\text{old}}}}(s_t^\pi)$ represents expected return of $\pi_{\theta_{\text{old}}}$ at $s_t$, $\hat{\Delta}_t$ is positive when the teacher trajectory earns a higher reward than the expected student return and negative vice-versa. Hence, the update to $\pi_\theta$ is relative to the student's value function over the actions generated by $\mu$. This is an extension of the advantage function for off-policy trajectories.

As Figure 2 shows, during the TADPO update, only the actor network and the feature encoder are updated using $L^{\text{TAD}}$ as in equation 8. The value function is only updated using trajectories generated by the student exploration process according to the intuition provided in 2.1.

Figure 3 gives an visualization of the value of distillation function as a function of $\rho$ and $\hat{\Delta}$. Using $\rho$ and $\hat{\Delta}$ in equation 8 ensures that the policy gradient only propagates when (i) the teacher trajectory rewards are higher than the expected student return and (ii) the student's likelihood of performing the same action $a_t$ is not significantly higher than that of the teacher. Also similar to PPO, $L^{\text{entropy}}$ in equation 8 regulates the exploration of the student policy.

A key observation is that in pure policy gradient methods, gradient computation means the policy improvement occurs only over a state distribution induced by the existing policy. However, it is both reasonable and feasible to improve the policy over other distributions. In particular, if one happened to already know the optimal distribution, or at least a better distribution, it could be advantageous to focus policy updates on that. Our modified algorithm does exactly that. Even if trajectories sampled

according to the existing (poor) student policy would be unlikely to visit some state $s_t$, $\pi(\cdot|s_t^\pi)$ can still be optimized using signals generated from the teacher trajectories.

## 3.2 TRAINING PROCEDURE

Teacher action distillation with policy gradients involves optimizing an actor function while simultaneously bootstrapping a value function with the expected return of that actor. Consequently, training a policy with TADPO requires interlacing trajectories sampled from both the teacher and student policies. Thus in on-policy settings, training the policy in separate phases of imitation learning and reinforcement learning does not yield a sufficiently accurate student value function, hindering effective learning from teacher trajectories.

We then propose algorithm 1 to enable the simultaneous training of the actor using teacher trajectories and student trajectories. In our implementation, $\hat{\Delta}_t$ is normalized to have standard deviation 1 in every mini-batch since the reward definition is unbounded.

---

**Algorithm 1** TADPO

1: **Input:** Teacher policy $\mu$, Student policy $\pi$, Teacher sample probability $p$
2: **Return:** Parameters of student policy $\theta$
3: Collect $N_\mu$ teacher transitions $\mathcal{B}_\mu \leftarrow \{\tau_{t_{a_t \sim \mu}} = (s_t^\mu, a_t, R_t, \mu(a_t|s_t^\mu))\}$
4: **for** iter = 1 to $I$ **do**
5:     Collect $N_\pi$ student transitions $\mathcal{B}_\pi \leftarrow \{\tau_{t_{a_t \sim \pi_{\theta_{\text{old}}}}} = (s_t^\pi, a_t, R_t, \pi_{\theta_{\text{old}}}(a_t|s_t^\pi))\}$
6:     **for** epoch = 1 to $K$ **do**
7:         **while** $\mathcal{B}_\pi \neq \emptyset$ **do**
8:             $r \sim \mathcal{U}(0, 1)$
9:             **if** $r > p$ **then**
10:                 Sample $n$ transitions $\tau \leftarrow \tau_t \sim \mathcal{B}_\pi$ without replacement
11:                 $\theta \leftarrow \text{PPOUpdate}(\tau)$
12:             **else**
13:                 Sample $n$ transitions $\tau \leftarrow \tau_t \sim \mathcal{B}_\mu$ without replacement
14:                 $\theta \leftarrow \text{TADPOUpdate}(\tau)$
15:             **end if**
16:         **end while**
17:         Reinitialize $B_\mu$ and $B_\pi$
18:     **end for**
19: **end for**

---

## 3.3 OFF-ROAD AUTONOMY STACK

As shown in Figure 1, we use two subsystems to achieve off-road autonomous driving: an MPPI-based high-level planner that generates waypoints towards a predefined goal using coarse, global information; and a RL-based controller that learns to track sparse waypoints using local information.

The MPPI planner for this problem is designed in accordance with Han et al. (2024), with the same cost function. For the teacher policy, MPPI provides dense waypoints and is referred further as MPPI-d. The teacher is a PPO controller trained to track provided MPPI-d waypoints, analogous to the hybrid controller for quadrupedal robots in Jenelten et al. (2024). During training and deployment of the student policy, MPPI provides sparse waypoints and is referred to as MPPI-s. Because of the high runtime cost of the MPPI-d planner, when generating teacher demonstrations, a fixed, pre-computed set of expert MPPI-d waypoints are used. By training with different waypoint distances, responsibility for planning at intermediate distances is shifted from the MPPI-d planner to the student policy controller. This allows for much faster and less frequent planning at deployment.

## 4 RESULTS AND DISCUSSIONS

### 4.1 EXPERIMENT SETUP

**Observation and Action Spaces**   We combine proprioceptive states with visual input for both the teacher and the student policies. The proprioceptive observation includes the vehicle's normalized speed, roll, pitch, and encodings of the current and next waypoint are provided, with the teacher using densely planned waypoints and the student using sparsely planned waypoints, as detailed in 4.1 and A.5.

The visual observations consist of top-down views for planning and a forward camera for obstacle avoidance, with the teacher policy using a local top-down image and the student policy employing a wider-area image with a lower spatial resolution. Both utilize a stack of 3 historical frames for the final observation. The controller directly outputs throttle and steering commands. More information about the observation and action spaces can be found in A.5 and A.6 respectively.

**Reward Function**   The reward function we designed for off-road navigation includes five key components. The progress reward encourages movement toward the goal by measuring the reduction in distance between the vehicle's current and previous positions, while collision and damage penalties address vehicle safety, a jerk penalty discourages sudden acceleration changes, and a success reward is granted for reaching waypoints. More details about reward functions can be found in A.9.

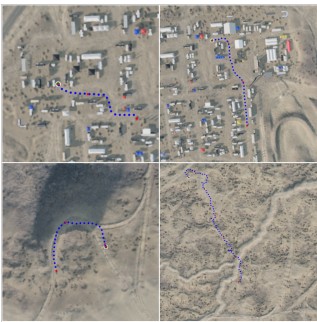 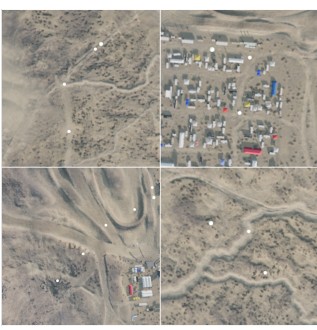 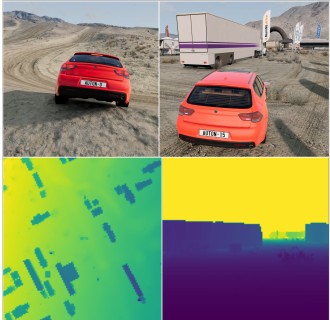

(a) Training set examples.          (b) Test set examples.          (c) Vehicle in the simulator.

Figure 4: Figure (a) shows example trajectories in the training set where the teacher policy is trained and demonstrations are collected. Blue dots are waypoints as observed by the teacher, and red dots are waypoints as observed by the student. Figure (b) shows examples from the test set, and only student waypoints are shown in white. The largest waypoint at the end of each trajectory is the goal. Both sets cover a diverse set of terrain that include obstacles at different scales, ditches, and cliffs. Figure (c) shows the vehicle running over a variety of terrain in simulation and two examples of camera views as observed by the controller.

**Training, Demonstration, and Testing Datasets**   We train both the teacher and the student over the same large map of a desert terrain. For any given start and goal position, we use a sparse global map and a global MPPI planner to generate waypoints of 80 m apart. We choose a fixed set of start, goal position pairs to serve as teacher training and demonstration trajectories. Using the MPPI planner supplied with a detailed local semantic segmentation map and associated depth information, we generate dense waypoints 6 m apart to span the intervals between the sparse 80m waypoints. A similar set is generated to serve as the test trajectories on which we evaluate all learned policies. For methods where expert labeling is required, we generate static, dense waypoints at the beginning of each episode.

We chose these trajectories to cover a range of offroad obstacles. As illustrated in Figure 4b, they can qualitatively be categorized as terrains with: i) positive obstacles, ii) extreme slopes, or iii) a hybrid of the preceding categories. The positive obstacles observed on the testing terrain mainly consist of natural obstacles (e.g. boulders and trees) and artificial obstacles (e.g. parked trailers, fences, etc.). Extreme slopes include ditches and sandy cliffs.

For teacher demonstrations, we collected 15 trajectories for (i) and (ii) each, and 20 for (iii). For evaluating models, we collected 8 for (i) and (ii) each, and 15 for (iii).

**Evaluation Metrics** We aggregate the evaluation performance of each policy over the test trajectories. For policies that produce an action distribution, we deterministically choose the mode of the distribution as the selected action. For each episode, we define the following metrics, with values normalized to the range [0,1]:

- Success Rate (`sr`): `sr` $= 1$ if the vehicle is within a completion radius $r$ of the goal position. `sr` $= 0$ otherwise.
- Completion Percentage (`cp`): `cp` measures the maximum progress the vehicle made towards the goal position, normalized by its initial distance to the goal position.
- Mean Speed (`ms`): `ms` is the mean speed of the vehicle during the episode.

More details about the metrics are included in A.7

**Simulator** We use `BeamNG` (BeamNG GmbH) as the simulator for our offroad vehicle. Visual example of the test vehicle driving in simulator is provided in Figure 4c. Further details regarding the simulator can be found in A.10.

## 4.2 RL AND IMITATION LEARNING BASELINES

| Controller | Planner | Extreme Slopes | | | Obstacles | | | Hybrid | | |
|---|---|---|---|---|---|---|---|---|---|---|
| | | `sr` | `cp` | `ms` | `sr` | `cp` | `ms` | `sr` | `cp` | `ms` |
| Teacher | MPPI-d | 0.88 | 0.94 | 5.83 | 1.00 | 1.00 | 5.91 | 0.94 | 0.96 | 5.69 |
| DAgger | MPPI-s | 0.00 | 0.58 | 1.96 | 0.00 | 0.83 | 1.62 | 0.00 | 0.79 | 1.68 |
| Vanilla PPO | MPPI-s | 0.00 | 0.14 | 0.38 | 0.00 | 0.25 | 0.49 | 0.00 | 0.37 | 0.40 |
| PPO+BC | MPPI-s | 0.00 | 0.25 | 0.94 | 0.00 | 0.40 | 0.78 | 0.00 | 0.32 | 0.84 |
| Vanilla SAC | MPPI-s | 0.00 | 0.01 | 1.71 | 0.00 | 0.16 | 1.64 | 0.00 | 0.24 | 1.61 |
| SAC+Teacher | MPPI-s | 0.00 | 0.50 | 1.21 | 0.00 | 0.29 | 1.24 | 0.00 | 0.58 | 1.24 |
| IQL | MPPI-s | 0.25 | 0.49 | 4.85 | 0.13 | 0.71 | 5.01 | 0.07 | 0.76 | 5.03 |
| TADPO[†] | MPPI-s[†] | **0.75** | **0.87** | **4.99** | **0.85** | **0.96** | **5.26** | **0.67** | **0.88** | **5.30** |

Table 1: Our method ([†]) compared with baselines, where `sr` denotes success rate, `cp` denotes average completion percentage, and `ms` denotes mean speed. MPPI-d refers to the local planner which outputs dense waypoints. MPPI-s refers to the global planner which outputs sparse waypoints. "Extreme Slopes" and "Obstacles" represent the challenging trajectories within the test set, while "Hybrid" refers to a combination of simpler and difficult trajectories. More information regarding the metrics is in A.7

Table 1 provides a comparison of our method with various RL and imitation learning baselines. Below, we briefly describe various intuitive and pre-existing RL baseline methods and their integration into our setup. All these policies that utilize teacher guidance are trained with same teacher trajectories using dense waypoint guidance from the MPPI planner. We also provide their quantitative performance in our test environment and discuss the reasons for any observed poor performance. We also provide their training reward curves in A.2.

### 4.2.1 IMITATION LEARNING METHODS

**DAgger** DAgger Ross et al. (2011) provides a straightforward method for supervising the policy by allowing queries to a teacher during training. Initially, the teacher trajectories are utilized for behavior cloning (BC) on the student. Subsequently, the student policy is enhanced by penalizing the discrepancy between the actions predicted by the student and those of the teacher at each state encountered. In complex environments like off-road driving, DAgger fails because of compounding errors. As the policy accumulates error and deviates from expert trajectories, it encounters unseen or irrecoverable states, thus severely degrading its performance.

### 4.2.2 ON-POLICY METHODS

**Vanilla PPO**  Vanilla PPO is trained as described in 2.1. This method does not utilize demonstrations from a teacher and is trained only on sparse waypoints. As described in 2.1, Vanilla PPO encounters exploration challenges in obstacle-rich terrains, which hinders its ability to learn an optimal strategy. Without guidance, the policy fails to differentiate between various types of terrains and defaults to a sub-optimal, overly-cautious strategy.

**PPO+BC**  A naive approach to distill teacher actions into the student is to incorporate a KL divergence loss between the predicted action distributions of the student and teacher. PPO+BC introduces a loss term that aligns the policy $\pi$ with the teacher policy $\mu$ across all encountered states. The vanilla PPO loss function is modified to $L_{\text{KL}}$ for training.

$$L^{\text{KL}} = L^{\text{PPO}} - \beta \text{KL}[\pi(a_t|s_t^{\pi}), \mu(a_t|s_t^{\mu})] \tag{12}$$

While this provides strong supervision, a issue similar to DAgger arises when the student queries the expert from out-of-distribution states and optimizes using sub-optimal action labeling. Additionally, the updates from the KL divergence term are unconstrained, which leads to unstable training and results in convergence to a sub-optimal policy.

### 4.2.3 OFF-POLICY METHODS

**Vanilla SAC**  SAC (Haarnoja et al. (2018)) is an off-policy RL algorithm that optimizes a stochastic policy and value function, enabling efficient and stable learning for continuous control tasks. SAC struggles in our high-exploration, multi-task setup because its entropy maximization can lead to excessive exploration of irrelevant actions, reducing its focus on task-specific objectives. This makes it less effective in environments requiring targeted exploration and adaptation to multiple tasks with distinct strategies.

**SAC+Teacher**  As an off-policy algorithm, SAC can utilize trajectories from teacher demonstrations without requiring any modifications to the algorithm. A portion of the replay buffer is pre-populated during the training. In this case, the buffer size remains consistent with TADPO, with the teacher trajectory ratio set at $p = 0.5$. As also shown in Yu et al. (2019), SAC does not perform well when it has to handle various different kinds of tasks (in this case, a very diverse set of terrains).

**IQL**  Implicit Q-learning (Kostrikov et al. (2021)) which extends Q-learning and actor critic methods is an off-policy reinforcement learning method that estimates Q-values without directly optimizing a policy, allowing the agent to implicitly select actions that maximize the value function. It is used in our teacher-student setup by following the actions suggested by the teacher's demonstrations, reinforcing behavior through the learning of Q-values associated with those actions. IQL demonstrates some success in navigating extreme slope terrains, but its overall performance does not match that of TADPO. As noted in Janner et al. (2022), IQL excels in single-task scenarios but faces challenges in multi-task environments, such as off-road autonomy. Given that off-road autonomy involves dynamically handling obstacle avoidance and rapid changes in how the vehicle and terrain interacts, IQL struggles to perform effectively in this setup.

### 4.3 MODEL-BASED BASELINES

The first section of Table 2 provides performance of the planner baselines for comparison. These planners are run while simulation is paused, allowing them to provide the next action before continuing. They show that with enough samples and planning horizon, these planners perform similarly well. The trained dense waypoint tracking policy, while following MPPI-d waypoints, perform similarly to a PID controller, but is more aggressive as indicated by its higher mean speed.

A key difference between these planners is their time of inference. We observe that inference time is more sensitive to $h$ than $N$, which reduces long-horizon understanding and, in turn, degrades real-time performance. Compared to MPPI, CEM takes a more iterative approach to sampling and evaluating action sequences, thus requires more compute time to plan. RL+MPPI enhances MPPI

by learning a terminal value function and a state-dependent action distribution, thereby reducing its required number of trajectories sampled and sampling horizon.

| Planner | Controller | Extreme Slopes | | | Obstacles | | | Hybrid | | | |
|---------|-----------|------|------|------|------|------|------|------|------|------|------|
| | | sr | cp | ms | sr | cp | ms | sr | cp | ms | ti |
| CEM-d | PID | 0.88 | 0.96 | 5.51 | 1.00 | 1.00 | 5.16 | 0.87 | 0.94 | 5.13 | 3.47 |
| MPPI-d | PID | 0.88 | 0.96 | 5.39 | 1.00 | 1.00 | 5.87 | 0.87 | 0.94 | 5.43 | 2.02 |
| RL+MPPI-d | PID | 0.88 | 0.96 | 5.26 | 1.00 | 1.00 | 5.88 | 0.87 | 0.94 | 5.40 | 1.77 |
| MPPI-d | Teacher | 0.88 | 0.94 | 5.83 | 1.00 | 1.00 | 5.91 | 0.94 | 0.96 | 5.69 | 2.02 |
| CEM-d* | PID | 0.38 | 0.49 | **5.52** | 0.25 | 0.38 | 5.16 | 0.27 | 0.43 | 5.13 | 0.13 |
| MPPI-d* | PID | 0.38 | 0.57 | 5.43 | 0.25 | 0.48 | **5.48** | 0.27 | 0.46 | 5.54 | 0.12 |
| RL+MPPI-d* | PID | 0.38 | 0.61 | 5.32 | 0.25 | 0.50 | 5.46 | 0.27 | 0.52 | **5.63** | 0.12 |
| MPPI-s$^{\dagger *}$ | TADPO$^{\dagger}$ | **0.75** | **0.87** | 4.99 | **0.85** | **0.96** | 5.26 | **0.67** | **0.88** | 5.30 | **0.002** |

Table 2: Our method ($^{\dagger}$) compared with baselines, where `sr` denotes Success Rate, `cp` denotes Completion Percentage, `ms` denotes mean speed, and `ti` is the Time of Inference for one control step. * denotes allotting a limited compute budget for main control loop necessary for real-time deployment. -d denotes planning at dense waypoint distances while -s denotes planning at sparse waypoint distances.

When running in real-time (as shown in the second section of Table 2), all three methods degrade drastically in performance because of a significantly reduced horizon $h$ and number of sampled trajectories $N$. This forces the planner to generate globally suboptimal waypoints, leading to worse performace. Because of the sparsity of the waypoints, MPPI-s can be run in parallel as the TADPO controller tracks the sparse waypoints. This enables MPPIs to select waypoints more efficiently and effectively, leading to significantly better real-time driving performance of MPPI-s+TADPO compared to other methods.

Hyperparamters are provided in A.3, and more details about MPPI implementation is provided in A.4.

## 4.4 TADPO

TADPO outperforms state-of-the-art RL baseline methods, demonstrating its ability to learn to navigate a diverse set of off-road terrains. The policy's success rate (`sr`) significantly surpasses that of other baseline methods. Additionally, TADPO attains a high mean speed (`ms`) across all test trajectory sets compared to all other controller baselines. For model-based baselines, MPPI-s+TADPO significantly outperforms all other planner baselines in real-time driving. The inference time of TADPO is notably lower than that of model-based methods, highlighting its effectiveness in environments with diverse terrains, where quick adaptive maneuvers are essential.

Through ablations we find that $\epsilon_{\mu} = 0.5$ and a constant $p = 0.5$ provides best performance of the algorithm which has been used for comparing with baselines. We include ablation studies with hyperparameters configurations in A.1.

## 5 CONCLUSION

We propose (i) a hierarchical off-road autonomy pipeline and (ii) a new hybrid policy optimization method TADPO. The pipeline combines the strengths of MPPI and RL to provide a robust solution for off-road autonomous driving in complex terrains. TADPO leverages a teacher-student paradigm with a novel policy gradient formulation to resolve the challenges of exploration and planning. Our experimental results demonstrate significant improvements in navigating challenging environments compared to existing RL and imitation learning methods, validating the potential of our approach. We plan to deploy this algorithm onto real vehicles in our future work. Source code for TADPO and videos of the pipeline in action are available at `https://github.com/tadpo-iclr/tadpo`.

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

## A APPENDIX

### A.1 HYPERPARAMETER ABLATIONS

We experiment with various hyperparameters in the TADPO implementation, using the best-performing values in our results.

We ablate on the ratio of teacher and student ratio $p$ with values 0.5, 0.7 and 0.3. and $\epsilon_\mu$ values 0.5, 0.3, 0.7.

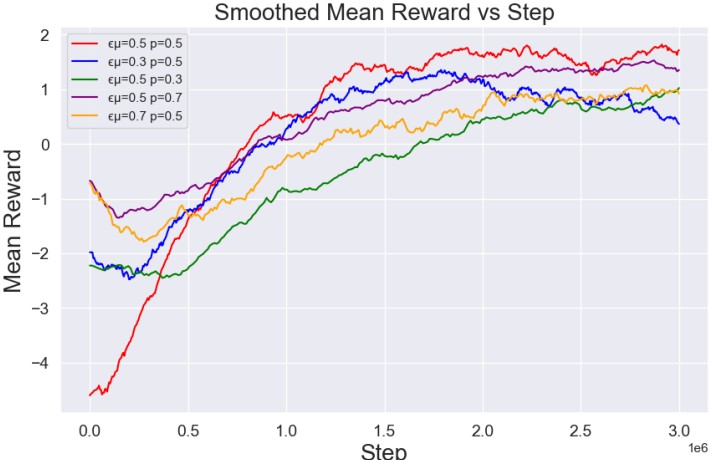

Figure 5: Smoothed mean reward

We find $p = 0.5$ and $\epsilon_\mu = 0.5$ to be optimal for our task.

## A.2 RL AND IMITATION LEARNING BASELINE TRAINING CURVES

We experimented with various RL and Imitation Learning baselines. We provide their training curves below.

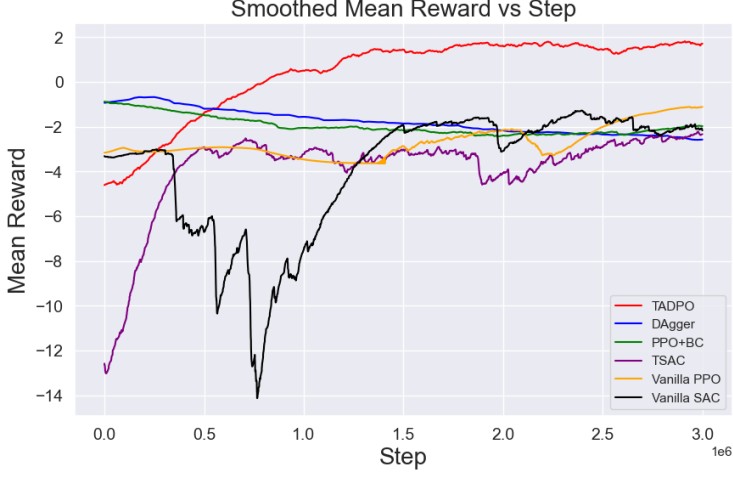

Figure 6: Smoothed mean reward comparison for baselines

## A.3 PLANNER HYPERPARAMETERS

| Planner | $N$ | $h$ | Number of iterations |
|---|---|---|---|
| MPPI-d | $1 \times 10^7$ | 30 | 1 |
| CEM-d | $1 \times 10^6$ | 30 | 3 |
| RL+MPPI-d | $1 \times 10^5$ | 20 | 1 |
| MPPI-d* | $1 \times 10^5$ | 4 | 1 |
| CEM-d* | $5 \times 10^4$ | 4 | 3 |
| RL+MPPI-d* | $1 \times 10^5$ | 4 | 1 |
| MPPI-s* | $1 \times 10^5$ | 4 | 1 |

Table 3: Comparison of Common Hyperparameters for Different Methods. * denotes alloting a limited compute budget for real-time deployment.

For training of RL+MPPI, the following hyperparameters are used.

| Hyperparameters | Value |
|---|---|
| Replay Buffer Size | $1 \times 10^5$ |
| Batch Size | 256 |
| Discount Factor ($\gamma$) | 0.99 |
| Critic Learning Rate | $1 \times 10^{-5}$ |
| Actor Learning Rate | $2 \times 10^{-5}$ |
| Learning Rate of $\alpha$ | $1 \times 10^{-5}$ |
| Model Learning Rate | $1 \times 10^{-4}$ |
| Expected Entropy ($\mathcal{H}$) | $\mathcal{H} = -4$ |
| Learning Rate for Target | 0.005 |
| Maximum Episode Length | 1000 |
| Training Iteration Number | $1 \times 10^6$ |

Table 4: RL+MPPI: RL Module Hyperparameters

A.4 MPPI PLANNER IMPLEMENTATION DETAILS

The MPPI Planner for this problem is designed in accordance with Han et al. (2024), employing the same model dynamics and cost function. The key difference lies in the waypoint sampling method: while Han et al. (2024) uses a fixed time-based sampling approach, our approach utilizes a fixed distance-based sampling method, where distance is a hyperparameter. The predicted trajectory rollout given by the bicycle kinematics model Kong et al. (2015). We use this fixed distance-based sampling to provide distance-based waypoints to the PPO policy. The algorithm takes in the final goal position ($g$) and outputs the waypoints $w_i$ for $i \in 1, .., h$ where $h$ is the planning horizon. It is important to note that MPPI uses a depth map to compute rollover and toppling costs, and an annotation map to calculate segmentation costs. The cost is calculated at time $t$ for all sampled $w_{j,i}$ where $j \in \{1, ..., N\}$ where $N$ is the number of samples.

The cost function is:

1. Goal cost: $u_1 * ||w_{j,i} - g||$

2. Rollover Cost: $u_2*$ Same as Han et al. (2024)

3. Toppling Cost: $u_3*$ Same as Han et al. (2024)

4. Segmentation Cost: $u_4 * sw_j$

5. Smoothness Cost: $u_5 * s_t^2$

and the weights $u_i$ for $i \in \{1, 5\}$ are re-tuned for optimal performance for our vehicle, being 1, 10, 10, 100, 0.8 respectively. Segmentation weights $sw_j$ for $j \in 1, ....5$ are obstacles = 1, rocks = 0.8, dirt = 0, sand = 0.2 and else = 0.

## A.5 OBSERVATION SPACE

The non-visual inputs to the teacher and student policies are the same except for the observed way-points.

At a time step $t$, given the environment state $s_t$, the vehicle has position $\mathbf{p}_t$, velocity $\mathbf{v}_t$, acceleration $\mathbf{a}_t$, roll $\theta_t$, pitch $\phi_t$, and yaw $\psi_t$ in Tait-Bryant format in the world frame $\mathbf{O}$.

For a planning horizon $h$, waypoint distance $d$, the MPPI planner generates $\mathbf{w}_{i,t} \in \mathbb{R}$ for $i \in \{1, ..., h\}$, where $d_2(\mathbf{w}_{i,t}, \mathbf{w}_{i+1,t}) = d$ for $i \in \{1, ..., h-1\}$ and $\mathbf{w}_{i,t}$ are in the vehicle frame. We also set $\mathbf{w}_{k,t} = \mathbf{w}_{h,t} \forall k > h$. The vehicle's relative yaw to waypoint $\mathbf{w}_{i,t}$ is $\beta_i = \arctan2(\mathbf{w}_{i,t}^0, \mathbf{w}_{i,t}^1)$, where $\mathbf{w}_{i,t}^j$ is the $j$-th element of $\mathbf{w}_{i,t}$ using 0-based indexing. The global planner plans with $d = 80$ from the starting position to goal position using a coarse global map. The local planner plans between waypoints generated by the global planner using detailed local maps.

At time step $t$, the environment maintains an index $i_t$ which indicates the next waypoint the vehicle should traverse through. If the vehicle is within some switching threshold distance $r_{\text{switch}}$, $i_{t+1} = i_t + 1$; otherwise, $i_{t+1} = i_t$. When $i_{t+1} > h$, we consider the traversal of the planned route successful.

We define the signed waypoint distance for time step $t$ and waypoint at index $i$ as

$$d_t^i = \begin{cases} d_2(\vec{p_t}, \vec{w}_{i,t}) & \text{if } \frac{\pi}{2} \leq \beta_t \leq \frac{3\pi}{2}, \\ -d_2(\vec{p_t}, \vec{w}_{i,t}) & \text{otherwise.} \end{cases} \tag{13}$$

The non-visual inputs to the policies then are $O_t = (d_t^{i_t}, d_t^{i_t+1}, \beta_{i_t}, \beta_{i_t+1}, \frac{|\vec{v_t}|}{v_{\max}}, \theta_r, \theta_p)$ where $v_{\max}$ is an arbitrary maximum speed of the vehicle being driven.

All observations made by the teacher and student are stacked with that generated by $s_{t-1}$ and $s_{t-2}$.

Also, we define $C_t^{\text{td}}(\text{rad}, \text{res}, \text{chan})$ to be a visual observation generated by a top-down camera at time $t$. The camera field-of-view is maintained so that when viewing flat ground at the level of the vehicle's center of mass, it would be able to observe a square with each side measuring $2 \cdot \text{rad}$ m. The camera's resolution is set to $(\text{res}, \text{res})$. $\text{chan}$ could be either $\text{rgbd}$, in which case a color image is stacked with a depth image, or $\text{depth}$, in which case only a depth image is presented. The camera uses a perspective camera model with a z-position of $240$ m above the vehicle.

Additionally, we define $C_t^{\text{f}}$ to be the observation of a front-facing camera at time $t$. The camera generates a $64 \times 64$ image in $\text{rgbd}$. It is positioned $(0.0, -1.5, 2.0)$ m offset from the center of mass of the vehicle. It has a field-of-view of $84°$.

### A.5.1 TEACHER POLICY

The teacher policy's observation at time $t$ is

$$s_t^\mu = (O_t^\mu, C_t^{\text{td}}(15, 64, \text{rgbd}), C_t^{\text{f}})$$

with $O_t^\mu$ generated by $r_{\text{switch}} = 3, d = 6$.

### A.5.2 STUDENT POLICY

The student policy's observation at time $t$ is

$$s_t^\pi = (O_t^\pi, C_t^{\text{td}}(30, 64, \text{rgbd}), C_t^{\text{td}}(90, 64, \text{depth}), C_t^{\text{f}})$$

with $O_t^\pi$ generated by $r_{\text{switch}} = 3, d = 80$.

## A.6 ACTION SPACE

The action space used is defined to be $(\tau_t, s_t)$ where $\tau_t$ is throttle and $s_t$ is the steering at time instance $t$. $s_t$ ranges from -1 (full right turn) to +1 (full left turn) and $\tau_t$ controls the gas pedal, with +1 for full forward acceleration and -1 for full reverse. Gear shifts are managed by the simulator's

controller, and brakes are applied when the vehicle's direction opposes $\tau_t$. Otherwise, $\tau_t$ controls the engine, moving the vehicle forward for positive values and backward for negative values.

## A.7 EVALUATION METRICS

For an episode, with vehicle position $\mathbf{p}_t$ and speed $v_t$ at time $t \in \{1...T\}$, goal position $\mathbf{p}_g$, acceptance radius $r_{\text{accept}}$, control period $\tau$, the evaluation metrics are given as follows:

$$\texttt{sr} = \begin{cases} 1 & d_2(\mathbf{p}_T, \mathbf{p}_g) < r_{\text{accept}} \\ 0 & \text{otherwise} \end{cases} \tag{14}$$

$$\texttt{cp} = 1 - \min_{t \in \{1...T\}} d_2(\mathbf{p}_t, \mathbf{p}_g) \tag{15}$$

$$\texttt{ms} = \frac{\sum_{i=1}^{T-1} d_2(\mathbf{p}_t, \mathbf{p}_{t+1})}{\tau \cdot T} \tag{16}$$

## A.8 HYPERPARAMETERS

### A.8.1 TEACHER POLICY (PPO)

| Hyperparameters | Value |
| --- | --- |
| Learning Rate | 3e-4 |
| Discount Factor ($\gamma$) | 0.99 |
| GAE Parameter ($\lambda$) | 0.95 |
| Clip Range ($\epsilon$) | 0.2 |
| Number of Epochs | 10 |
| Mini-batch Size | 256 |
| Number of Steps per Update | 2048 |
| Value Function Coefficient ($\lambda_v$) | 0.5 |
| Entropy Coefficient ($\lambda_e$) | 0.001 |
| MLP Network Architecture | [128,64,64] |
| CNN Feature Extractor | NatureCNN (Mnih et al. (2013)) |
| CNN Latent Space | 256 |

Table 5: Hyperparameters for Teacher and Student Training

### A.8.2 STUDENT POLICY (TADPO)

The PPO part of the student is trained using the same hyperparameters as in A.8.1. Hyperparameters for the TADPO update are as follows.

| Hyperparameters | Value |
| --- | --- |
| Update ratio ($\epsilon_\mu$) | 0.5 |
| Teacher policy ratio ($p$) | 0.5 |
| Learning Rate | 3e-4 |
| Discount Factor ($\gamma$) | 0.99 |
| Clip Range ($\epsilon$) | 0.2 |
| Number of Epochs | 20 |
| Mini-batch Size | 256 |
| Number of Steps per Update | 2048 |
| Value Function Coefficient ($\lambda_v$) | 0.5 |
| Entropy Coefficient ($\lambda_e$) | 0.001 |
| MLP Network Architecture | [128,64,64] |
| CNN Feature Extractor | NatureCNN (Mnih et al. (2013)) |
| CNN Latent Space | 256 |
| Teacher Demonstration Buffer Size | 1e5 |

Table 6: Hyperparameters for Teacher and Student Training

## A.9 REWARDS

The reward function is designed to encourage progress towards the desired waypoint at $t$, while penalizing collisions, excessive jerk, and vehicle damage. Additionally, a success reward is granted upon reaching the final waypoint.

1. Progress: $c_1 * (||\vec{p_{t-1}} - \vec{w_{i,t}}|| - ||\vec{p_t} - \vec{w_{i,t}}||)$

2. Collision: $\begin{cases} c_2 & \text{if } \text{dam} > \text{dam}_{\text{thresh}} \\ 0 & \text{otherwise} \end{cases}$

3. Damage: $c_3 * \text{dam}$

4. Jerk: $c_4 * (||a_t - a_{t-1}||/dt)$

5. Success: $\begin{cases} c_5 & \text{if } ||\vec{p_{t-1}} - \vec{w_{i,t}}|| < w_{\text{thresh}} \\ 0 & \text{otherwise} \end{cases}$

where $c_i$ for $i \in \{1, \ldots, 5\}$ are scaling factors for the rewards, with values of 1, -2, -1, -0.003, and 1, respectively. The progress reward reflects the distance the vehicle travels toward the goal, with the maximum reward between two waypoints being equal to the distance between them.

These rewards are significantly sparse for exploration in the off-road navigation problem that involve navigating diverse terrains and obstacles.

## A.10 SIMULATOR

### A.10.1 BEAMNG

We use BeamNG.tech (BeamNG GmbH) as the simulator for training and evaluating our algorithms. BeamNG.tech offers a highly realistic simulation environment, featuring advanced vehicle dynamics and damage modeling. BeamNG.tech offers detailed vehicle dynamics and damage modeling, allowing us to test our algorithms in a realistic environment that closely mirrors real-world conditions.

We use `etk800` as our vehicle. The car has dimensions of 2 meters in width, 4.7 meters in length, and 1.4 meters in height. It features a simulated internal combustion engine, an automatic transmission, and an artificially imposed speed limit of $30m/s$.

## A.11 ALGORITHM IMPLEMENTATIONS

We use existing software packages for the implementations of the baselines. We use existing implementation of MPPI by https://github.com/UM-ARM-Lab/pytorch_mppi and CEM by https://github.com/LemonPi/pytorch_cem for our planners. We use the official TD-MPC implementation by Hansen et al. (2022). We use the DAgger implementation included in `imitation` by Gleave et al. (2022). We also use the official implementation of IQL (Kostrikov et al. (2021)). For SAC and PPO, we use Stable Baselines 3 (`SB3`) by Raffin et al. (2021). We implement our algorithm and other baseline algorithms based on the `SB3` framework. We publish the source code for our method at https://github.com/tadpo-iclr/tadpo.

