# OpenReview forum: "Towards Off-Road Autonomous Driving via Planner Guided Policy Optimization"
_ICLR.cc/2025/Conference — Submitted to ICLR 2025_

### Official Review · Reviewer_1vog · 2024-10-30

**Soundness:** 3
**Presentation:** 3
**Contribution:** 2
**Rating:** 6
**Confidence:** 3

**Summary:**

The paper presents a novel approach for off-road autonomous driving by integrating a Model Predictive Path Integral (MPPI) planner with a reinforcement learning (RL) controller. The authors propose a teacher-student framework to address exploration challenges in RL, utilizing a new policy gradient formulation called Teacher Action Distillation with Policy Optimization (TADPO) extended from Proximal Policy Optimization (PPO).

The contribution of this paper can be summarized as follows:

1. The authors combine MPPI and RL in a teacher-student framework to handle long-horizon planning and real-time execution in challenging terrains.

2. The authors extend Proximal Policy Optimization (PPO) with Teacher Action Distillation (TADPO) to leverage off-policy trajectories for better training efficiency.

3. The authors conduct experiments on an offroad driving simulator to demonstrate the advantage of their proposed method over other RL and imitation learning methods.

**Strengths:**

The strengths of the paper are twofold.

1. The integration of MPPI with RL in a teacher-student framework to tackle off-road driving challenges is innovative. The hierarchical framework is well-motivated and addresses key limitations of both techniques.

2. The experiments conducted in a realistic off-road simulator demonstrate the effectiveness of the proposed method compared to various baselines, highlighting its potential for real-world applications.

**Weaknesses:**

The weaknesses of the paper is as follows:

1. **Selection of Baseline Methods:** The selection of baseline methods is not comprehensive. The baselines consist solely of learning-based planning approaches, specifically imitation learning and reinforcement learning. As stated by Dauner et al. in their CoRL 2023 paper, *Parting with misconceptions about learning-based vehicle motion planning*, traditional planning methods often outperform many learning-based planning methods in on-road autonomous driving. This conclusion may also apply to off-road scenarios. Therefore, it would be beneficial to include one or two traditional planning methods as baselines, such as the MPPI algorithm mentioned in the paper, or other Model Predictive Control (MPC) algorithms or sampling-based motion planning techniques.

2. **Complexity Analysis:** A complexity analysis of the proposed method is necessary, as efficiency is one of the advantages claimed by the authors. Specifically, the paper should include the inference time of the proposed method and a comparison with the baseline methods to demonstrate that the authors' approach is capable of real-time planning and is more efficient than previous approaches.

**Questions:**

In addition to the questions raised in the weaknesses, I have one more question regarding the rationale for choosing MPPI. Is MPPI the only state-of-the-art (SOTA) traditional planning method for off-road autonomous driving? If that is the case, it would be helpful for the authors to clarify this in the introduction. If there are other methods available, a brief review of existing SOTA traditional planning methods for off-road autonomous driving could provide valuable context, along with an explanation of why MPPI was chosen to be combined with reinforcement learning.

---

> ### Author Response · Authors · 2024-11-23
>
> Thanks for your valuable feedback and suggestions. We appreciate the effort and time invested in evaluating our submission. We will address the comments below.
>
> > Selection of Baseline Methods: The selection of baseline methods is not comprehensive. The baselines consist solely of learning-based planning approaches, specifically imitation learning and reinforcement learning. As stated by Dauner et al. in their CoRL 2023 paper, Parting with misconceptions about learning-based vehicle motion planning, traditional planning methods often outperform many learning-based planning methods in on-road autonomous driving. This conclusion may also apply to off-road scenarios. Therefore, it would be beneficial to include one or two traditional planning methods as baselines, such as the MPPI algorithm mentioned in the paper, or other Model Predictive Control (MPC) algorithms or sampling-based motion planning techniques.
>
> Thanks for the suggestion. Based on your comments, we have added CEM and RL+MPPI as baseline methods to our paper. They can also be found in Table 2 (as shown in the updated paper and Summary of Changes).
>
> > Complexity Analysis: A complexity analysis of the proposed method is necessary, as efficiency is one of the advantages claimed by the authors. Specifically, the paper should include the inference time of the proposed method and a comparison with the baseline methods to demonstrate that the authors' approach is capable of real-time planning and is more efficient than previous approaches.
>
> Thanks for the suggestion. We have provided the time of inference comparisons with all the algorithms in the paper. They can also be found in Table 2.
>
> > In addition to the questions raised in the weaknesses, I have one more question regarding the rationale for choosing MPPI. Is MPPI the only state-of-the-art (SOTA) traditional planning method for off-road autonomous driving? If that is the case, it would be helpful for the authors to clarify this in the introduction. If there are other methods available, a brief review of existing SOTA traditional planning methods for off-road autonomous driving could provide valuable context, along with an explanation of why MPPI was chosen to be combined with reinforcement learning.
>
> Yes, MPPI is currently the sole state-of-the-art (SOTA) method for planning in off-road autonomous driving. MPPI is a sampling-based method that applies importance-weighted optimization to generate control outputs. We have also implemented CEM and RL+MPPI and chose MPPI due to its high performance and ease of implementation. This is mentioned in section 2.2 and we have modified our introduction to clarify this.
>
> We hope our response clarifies the concerns raised and demonstrates the contributions of our paper. Thank you for your time and consideration.

---

> > ### Comment · Reviewer_1vog · 2024-12-03
> >
> > Thanks for your detailed response. I will keep my positive recommendation.

---

### Official Review · Reviewer_78ta · 2024-10-31

**Soundness:** 3
**Presentation:** 3
**Contribution:** 2
**Rating:** 3
**Confidence:** 3

**Summary:**

This paper proposes a hierarchical approach for off-road autonomous driving that combines model predictive path integral (MPPI) and reinforcement learning (RL). The high-level MPPI planning provides dense waypoints for exploration, while the low-level RL learns to navigate based on both visual and proprioceptive inputs. Extensive experiments in BeamNG simulations show the approach’s improved performance over several baseline RL and imitation learning methods especially in obstacle-dense, off-road terrains.

However, TADPO’s reliance on dense teacher guidance raises concerns about generalizability, particularly in dynamic or unfamiliar environments where such detailed waypoints may be unavailable. Additionally, while the model performs well in simulations, its adaptability to real-world conditions with diverse terrains, unpredictable weather, and lighting changes remains untested and uncertain. Broader comparisons with end-to-end DRL models could also help clarify the strengths and limitations of the hierarchical framework, especially for real-world applications.

**Strengths:**

The combination of high-level MPPI planning with a low-level RL controller creates an efficient approach for off-road autonomous driving.

The low latency of the RL controller allows for real-time decision-making, making the framework well-suited for off-road applications.

**Weaknesses:**

**Generalization to Real-World Conditions**: This approach has shown strong performance in simulation, but its ability to generalize to real-world off-road environments with diverse terrains and unpredictable elements is unclear. Real-world variability, such as changes in weather, lighting, and unexpected obstacles, could significantly impact its effectiveness.

**Reliance on Dense Teacher Guidance**: The teacher-student setup assumes access to densely planned waypoints for training, which may not be feasible in all scenarios. This dependency could hinder generalizability, especially in dynamically changing or unknown environments.

**Exploration Limitations**: While TADPO leverages the MPPI teacher for exploration guidance, the DRL student policy may still face challenges with effective exploration in complex terrain. The end-to-end navigation policy has limited learning benefits from the MPPI teacher, making it unclear if the proposed method resolves the exploration limitations inherent to PPO.

**Questions:**

**Ensuring Safety in High-Risk Environments**: Since TADPO uses an end-to-end DRL approach, how does it ensure safe navigation, especially in challenging off-road conditions where mistakes could lead to accidents or costly damage? What specific measures are in place to guarantee safety throughout the process?

---

> ### Author Response · Authors · 2024-11-23
>
> Thanks for your valuable feedback and suggestions. We appreciate the effort and time invested in evaluating our submission. We will address the comments below.
>
> > Generalization to Real-World Conditions: This approach has shown strong performance in simulation, but its ability to generalize to real-world off-road environments with diverse terrains and unpredictable elements is unclear. Real-world variability, such as changes in weather, lighting, and unexpected obstacles, could significantly impact its effectiveness
>
> We did not mean to imply that we had evidence to show we cannot handle domain shifts. We were only observing that we have not yet demonstrated this ability by deploying on a real vehicle. We are planning to deploy it in the future. We have also revised the conclusion section of the paper to clarify this.
>
>
> > Reliance on Dense Teacher Guidance: The teacher-student setup assumes access to densely planned waypoints for training, which may not be feasible in all scenarios. This dependency could hinder generalizability, especially in dynamically changing or unknown environments.
>
> We believe that dense teacher guidance is available in all scenarios as the teacher guidance is collected in simulation on a fixed set of trajectories. However, the student is trained on randomized start and goal positions over which teacher guidance is not necessary. Hence, it does not provide hindrance to generalizability.
>
> > Exploration Limitations: While TADPO leverages the MPPI teacher for exploration guidance, the DRL student policy may still face challenges with effective exploration in complex terrain. The end-to-end navigation policy has limited learning benefits from the MPPI teacher, making it unclear if the proposed method resolves the exploration limitations inherent to PPO.
>
> We understand the concern about exploration in complex terrains. However, we believe that guidance is always available within the simulation environment. The random initialization of start and goal points leads to a diverse distribution of terrains which, combined with guidance from the MPPI teacher, enables effective exploration in TADPO.
> As demonstrated by the reward curves in Figure 6, without guidance from the MPPI teacher, Vanilla PPO struggles to learn the desired planning behavior. We attribute the successful learning of the planning behavior to the effective guidance provided by MPPI, which helps overcome the exploration limitations inherent in PPO.
>
> > Ensuring Safety in High-Risk Environments: Since TADPO uses an end-to-end DRL approach, how does it ensure safe navigation, especially in challenging off-road conditions where mistakes could lead to accidents or costly damage? What specific measures are in place to guarantee safety throughout the process?
>
> Since TADPO has only been deployed in simulation, we have not yet implemented safety systems for guaranteeing vehicle integrity in real-world conditions. During training, we penalize damage, which helps the agent avoid obstacles effectively and achieve performance comparable to that of an MPC controller.
>
> For deployment on a real vehicle, our target vehicle has an auxiliary safety monitoring system that will run alongside the main control algorithm. This system will perform preventative maneuvers to ensure safe navigation and protect the vehicle from potential damage during operation.
>
> We hope our response clarifies the concerns raised and demonstrates the contributions of our paper. Thank you for your time and consideration.

---

### Official Review · Reviewer_hN5d · 2024-11-04

**Soundness:** 1
**Presentation:** 1
**Contribution:** 2
**Rating:** 1
**Confidence:** 4

**Summary:**

In this paper, the authors consider a hierarchical framework of a sparse global MPPI planner and a dense local MPPI planner setting waypoints tracked by low-level RL control policies. As the dense local planner has high inference costs, the approach tries to amortize the computational costs. An objective called TADPO, which is a reformulated PPO object, distills the knowledge of a teacher, who has access to the dense planner, to a student, who has only access to the sparse planner. The empirical results show that the sparse MPPI planner combined with the low-level student policy performs significantly better than RL or IL methods learned from scratch.

**Strengths:**

- Novel Idea and Setup: The idea of distilling the teacher into the student via PPO is interesting and the problem of off-road driving seems to be challenging.

**Weaknesses:**

- Correctness: The description of MPPI seems to be wrong, different to eq. (7), MPPI uses importance sampling to average between all sampled trajectories.
- Showing the stated claims: From my understanding the experiments in Table 1 do not validate the modified PPO objective over a simple BC loss. Apart from MPPI + Teacher and TADPO all other methods seem to not use the sparse MPPI planner during testing. Thus, in that sense, only the tracking formulation with the MPPI planner is validated, but not whether training the student requires the TADPO objective and a simple BC loss would not work.
- Missing related work (or even important baselines): A comparison to [1] would be interesting where MPPI is accelerated by rolling out trajectories with an RL policy. I think this idea should be compared to the algorithm presented in this work. Another direct competitor could be variants where the terminal value function is learned, thus significantly improving the convergence of MPPI [1], [5], [6].  Another research line very related to the objective presented in this work is guided policy search as defined in [2].  Also, a potentially important related paper could be [3] where a reference generated by a high-lvl MPC planner is tracked by a low level RL policy.
- Writing: The description and notation of the algorithm could be improved. It is unclear to me which dynamics model the authors are using for the sparse MPPI or the dense MPPI planner. Further, it was not completely clear, whether the baselines use the MPPI planner or not.

[1] Y. Qu et al, "RL-Driven MPPI: Accelerating Online Control Laws Calculation With Offline Policy," in IEEE Transactions on Intelligent Vehicles, vol. 9, no. 2, pp. 3605-3616, Feb. 2024.
[2] S. Levine and V. Koltun, "Guided Policy Search," in Proceedings of the 30th International Conference on Machine Learning, 2013.
[3] F. Jenelten et al, "DTC: Deep Tracking Control—A Unifying Approach to Model-Based Planning and Reinforcement Learning for Versatile and Robust Locomotion," arXiv preprint arXiv:2309.15462, Jan. 2024.
[5] N. A. Hansen, H. Su and X. Wang, "Temporal Difference Learning for Model Predictive Control," in Proceedings of the 39th International Conference on Machine Learning, 2022.
[6] K. Lowrey  et al, "Plan Online, Learn Offline: Efficient Learning and Exploration via Model-Based Control", in Proceedings of the 7th International Conference on Learning Representations, 2019.

**Questions:**

- Instead of greedily selecting the most cost-efficient trajectory, one could average via the importance sampling scores like in MPPI or use a Cross-Entropy Planner [4]. Do you think this would help the planner to converge more stably?

[4] M. Kobilarov, "Cross-entropy Motion Planning," The International Journal of Robotics Research, vol. 31, no. 7, pp. 855-871, 2012.

Overall:
- The paper would benefit from more clarity: One should state more clearly what the novel key algorithmic contributions are and why they offer an improvement over the related work mentioned above. I further think that a more detailed analysis of the altered PPO objective would significantly improve the paper!

Further Remarks:
- Notation in equation (1) does not make sense, on the right side the argmax returns a function, and on the left side, a probability density is evaluated.
- Are you using the generalised advantage estimator for PPO. In the Appendix there is a hyperparameter suggesting so, but from the notation in the main paper it was not clear to me !?
- The introduction is rather lengthy and would benefit from shortening.
- Figure 5 scaling is off.
- On page 9, a formulation in the paragraph SAC+Teacher describes putting the trajectories into the replay buffer as being simplistic. However, it is not described whether this approach was used in the end.
- In Figure 1, the proposed method is called MPPI + TADPO, whereas in Table 1, it is called TADPO. This could lead to potential confusion.
- In Appendix A.3.2 it should be a student policy, not a teacher policy.
- There are a lot of figures not cited in the text: Figure 2, 3 and 5.

---

> ### Author Response · Authors · 2024-11-23
>
> Thanks for your valuable feedback and suggestions. We appreciate the effort and time invested in evaluating our submission. We will address the comments in the following replies.
>
> > The paper would benefit from more clarity: One should state more clearly what the novel key algorithmic contributions are and why they offer an improvement over the related work mentioned above. I further think that a more detailed analysis of the altered PPO objective would significantly improve the paper!
>
> Thanks for your comments! We have made some revisions to the paper to enhance clarity. To summarize, our key algorithmic contributions are twofold: (i) we created a novel method to train modified PPO (i.e. TADPO) RL policies using teacher demonstrations; and (ii) we combined TADPO with a sparse MPPI planner to achieve real-time off-road driving. Combining these algorithmic contributions, we demonstrate that RL algorithms can be used to effectively learn planning and work synergistically with model-based planning techniques.
>
> As for a more detailed analysis of TADPO, we included a lot of intuition in section 2.1 and 3.1. Specifically, section 2.1 analyzes the existing PPO equations to motivate TADPO, and 3.1 gives a detailed analysis of why TADPO might perform well intuitively. If there are any particular concerns about clarity in these sections or any other suggestions, we are happy to provide more information.
>
> > Correctness: The description of MPPI seems to be wrong, different to eq. (7), MPPI uses importance sampling to average between all sampled trajectories.
>
> Thanks for pointing it out. Eq. (7) was intended to depict the general Model Predictive Control framework. In the updated description, we describe MPC as a general control framework to generate cost-minimizing trajectories over a given horizon. MPPI, a sampling-based MPC technique, utilizes importance-weighted optimization for efficient control generation, offering advantages in parallelizability and speed compared to other traditional sampling-based MPC techniques. We have clarified our description of MPC and MPPI in section 2.2.
>
> > Showing the stated claims: From my understanding the experiments in Table 1 do not validate the modified PPO objective over a simple BC loss. Apart from MPPI + Teacher and TADPO all other methods seem to not use the sparse MPPI planner during testing. Thus, in that sense, only the tracking formulation with the MPPI planner is validated, but not whether training the student requires the TADPO objective and a simple BC loss would not work.
>
> Teacher uses dense MPPI waypoints while all other listed methods in Table 1 use sparse MPPI waypoints for both training and testing. We have updated Table 1 and 2. Both now have separate columns indicating the planner and controller used by each algorithm. We also mention in Section 4.2 how we have utilized the teacher guidance in the baselines (if applicable). Thus, we believe the listed algorithms serve as sufficient baselines to TADPO and validates its claimed effectiveness. In particular, our algorithm against DAgger is a good illustration of our modified algorithm performing better than a simple BC loss. As an additional experiment, we also trained a controller policy with a simple BC loss and sparse MPPI waypoints. It shows poor performance as shown in Table 3.

---

> ### Author Response · Authors · 2024-11-23
>
> > Missing related work (or even important baselines): A comparison to [1] would be interesting where MPPI is accelerated by rolling out trajectories with an RL policy. I think this idea should be compared to the algorithm presented in this work. Another direct competitor could be variants where the terminal value function is learned, thus significantly improving the convergence of MPPI [1], [5], [6]. Another research line very related to the objective presented in this work is guided policy search as defined in [2]. Also, a potentially important related paper could be [3] where a reference generated by a high-lvl MPC planner is tracked by a low level RL policy.
>
> > Instead of greedily selecting the most cost-efficient trajectory, one could average via the importance sampling scores like in MPPI or use a Cross-Entropy Planner [4]. Do you think this would help the planner to converge more stably?
>
> We have cited [1] RL+MPPI, [3] DTC, [4] CEM, and [5] TD-MPC. In particular, we note that DTC's training procedure is very analogous to how our Teacher policy is trained. We have also incorporated RL+MPPI and CEM into our baseline comparisons, shown in Table 2. An updated Table 2 is also provided in the Summary of Changes for your reference.
>
> We have performed additional experiments for [2] GPS and [5] TD-MPC. Their results are provided in Table 3.
>
> [2] Guided Policy Search (GPS) tries to optimize a controller and a policy learned by supervised learning simultaneously. The controller has more state information (dense waypoints in our case) compared to the policy which has less state information (sparse waypoints). We train the controller in the first phase, and in the second phase improve the controller, while using the sampled trajectories to train the policy using supervised learning. The results for these are provided in Table 3. This method performs poorly and seems quite similar to DAgger when given a trained expert policy.
>
> [5] TD-MPC attempts to learn the control task from scratch without any explicitly embedded priors, which severely limits its effectiveness in terms of learning to plan. The results in Table 3 show that it was not effective for this task even when not enforcing the real-time compute constraint. Additionally, many of its key ideas to improve convergence of MPPI were also incorporated in RL+MPPI, so we feel RL+MPPI is sufficient to serve as a baseline.
>
> [6] POLO describes an exploration technique to learn value functions when given a perfect world model. However, this is a significantly different setup than the off-road driving task where the key constraint lies in the MPC sampling speed. Similar to TD-MPC, the core ideas for ensuring faster convergence of the planner are also present in RL+MPPI. Therefore, we did not include it in the updated version of the paper.
>
> > Writing: The description and notation of the algorithm could be improved. It is unclear to me which dynamics model the authors are using for the sparse MPPI or the dense MPPI planner. Further, it was not completely clear, whether the baselines use the MPPI planner or not.
>
> As pointed out earlier in this reply, we have updated both Table 1 and 2 to clarify which planner and controller each algorithm uses. As mentioned in Appendix Section A.4, we use the same setup as by Han et al. in (Model predictive control for aggressive driving over uneven terrain) for both sparse and dense MPPI planners, including the model to compute the rollouts. The predicted trajectory rollout is given by the bicycle kinematics model [7].
>
> [7] J. Kong, M. Pfeiffer, G. Schildbach and F. Borrelli, "Kinematic and dynamic vehicle models for autonomous driving control design," 2015 IEEE Intelligent Vehicles Symposium (IV), Seoul, Korea (South), 2015, pp. 1094-1099.
>
> > Are you using the generalised advantage estimator for PPO. In the Appendix there is a hyperparameter suggesting so, but from the notation in the main paper it was not clear to me !?
>
> For the TADPO method, we are not using GAE because of the computational complexity of reevaluating the updated value functions over the entire teacher replay buffer states when the policy value function is updated. GAE is used, however, in the on-policy learning phase. We have updated the paper to reflect this fact.
>
> > On page 9, a formulation in the paragraph SAC+Teacher describes putting the trajectories into the replay buffer as being simplistic. However, it is not described whether this approach was used in the end.
>
> We pre-populate 50% of the replay buffer with teacher rollouts in the implementation of SAC+Teacher. We have updated the paper to improve clarity.
>
> We have updated the manuscript based on the feedback provided, and we believe these changes have improved the clarity and robustness of our work. We hope our responses clarify the concerns raised and demonstrate the contributions of our paper. Thank you for your time and consideration.

---

> ### Author Response · Authors · 2024-11-23
>
> ### Table 3 (Additional Experiments)
>
> | Ref      | Planner  | Controller            | Extreme Slopes | Extreme Slopes | Extreme Slopes | Obstacles | Obstacles | Obstacles | Hybrid   | Hybrid   | Hybrid   |
> | -------- | -------- | --------------------- | -------------- | -------------- | -------------- | --------- | --------- | --------- | -------- | -------- | -------- |
> |          |          |                       | `sr`           | `cp`           | `ms`           | `sr`      | `cp`      | `ms`      | `sr`     | `cp`     | `ms`     |
> |     | MPPI-s   | Behavior Cloning(BC) | 0.00  | 0.19  | 2.34 | 0.00 | 0.32 | 3.21| 0.00 | 0.23 | 3.46 |
> | **[2]**  | MPPI-s   | Guided Policy Search  | 0.00           | 0.46           | 2.43           | 0.00      | 0.44      | 3.44      | 0.00     | 0.47     | 3.59     |
> | **[5]**  | TD-MPC-d | PID                   | 0.00           | 0.24           | 0.34           | 0.00      | 0.54      | 0.23      | 0.00     | 0.28     | 0.12     |
> |          | MPPI-s   | TD-MPC                | 0.00           | 0.35           | 3.64           | 0.00      | 0.21      | 4.13      | 0.00     | 0.43     | 3.95     |
> | **Ours** | MPPI-s   | TADPO                 | **0.75**       | **0.87**       | **4.99**       | **0.85**  | **0.96**  | **5.26**  | **0.67** | **0.88** | **5.30** |
>
> In this table, sr denotes success rate, cp denotes average completion percentage, and ms denotes mean speed. MPPI-d refers to the local planner which outputs dense waypoints. MPPI-s refers to the global planner which outputs sparse waypoints. “Extreme Slopes” and “Obstacles” represent the challenging trajectories within the test set, while “Hybrid” refers to a combination of simpler and difficult trajectories. More information regarding the metrics is in A.7.

---

> ### Author Response · Authors · 2024-12-04
>
> Dear Reviewer hN5d,
>
> We kindly and respectfully disagree with the "1: Strong Reject" rating given to our paper. We believe that we have thoroughly addressed the concerns and questions raised, and we sincerely feel that the rating could be reconsidered and adjusted based on the clarifications provided.
>
> We would like to respectfully clarify that the primary concern raised by the reviewer regarding the necessity of our framework stems from the belief that the baselines do not employ the same hierarchical approach as our proposed method (using sparse MPPI waypoints - MPPI-s). This is a misunderstanding, as we have addressed this point in our rebuttal (Rebuttal Reply 1). We also demonstrate that a simple BC loss, along with various other RL and Imitation learning methods, perform poorly compared to our method. Hence, we firmly believe that our method of distilling Teacher actions to a student (TADPO) is not only novel but also a crucial advancement that addresses significant gaps in existing methods.
>
> We have also addressed all the additional baselines (RL+MPPI, DTC, TD-MPC, CEM) that the reviewer suggested as potential challengers to our approach. While some of these baselines were not directly relevant to our problem, we adapted them to our problem and provided meaningful results and intuition for the same.
>
> If the reviewer still feels that there are any remaining gaps or unclear aspects in the paper, we would greatly appreciate the opportunity to address them and provide further clarification in response to our rebuttal.
>
> In conclusion, we believe that this work represents a significant contribution to the advancement of the field of learning for autonomy, and we respectfully encourage the reviewer to reconsider their rating.
>
> We sincerely appreciate the time and effort invested in reviewing our paper and are grateful for the opportunity to engage in this constructive discussion.

---

### Official Review · Reviewer_BkEv · 2024-11-04

**Soundness:** 3
**Presentation:** 3
**Contribution:** 3
**Rating:** 6
**Confidence:** 3

**Summary:**

The paper presents an approach to learn policies for off-road autonomous driving on diverse and challenging terrains. The authors try to tackle the problems of model-based control (MPPI) only approaches and RL-only based approaches and combine the two methodologies to get a low-frequency high-level MPPI planner and a high-frequency low level RL controller.

They extend the PPO (proximal policy optimization) algorithm to enhance targeted exploration. They introduce a teacher-student paradigm where the planning information is distilled from the teacher policy (trained using dense planning information from MPPI) and the student policy learns the off-road traversal. They call this extended algorithm Teacher Action Distillation Policy Optimization (TADPO).

The paper compares this algorithm with other on-policy and off-policy algorithms using success rate, completion percentage and mean speed and the evaluation metrics. TADPO outperforms SOTA RL baselines methods in terms of learning to navigate in a diverse set of off-road terrains.

**Strengths:**

- The authors did a good elaborate comparison of why some of the existing algorithms (IL, vanilla PPO, on-policy, off-policy) might not be good for traversal in diverse terrains that helps in getting a good contrastive contribution of the presented work.

- The algorithm helps in improving the policy (in policy gradient methods) over other better state distributions (using the teacher demonstrated trajectories).

- The framework allows for faster and less frequent planning at the time of deployment.

- The paper designs the algorithm to make it capable of real-time execution.

**Weaknesses:**

- The pipeline is not able to adapt to different domains, hence generalizability is a problem for now.

- The work seems to be not ready for real-world deployment and is best suited for simulation based research.

- The authors have presented why other IL methods, on-policy methods and off-policy methods might not perform well for off-road terrain considering the diverse scenarios; but it would be good to have some contrastive quantitative metrics to present how TADPO is trying to resolve those challenges, for example handling diverse scenes.

**Questions:**

- Nit: Line 141: There is no A_cap in equation 2.

- Nit: Line 704: β_{i} = (w^{0}_{i,t}, w^{1}_{i,t}) - here the second `w` can be made bold?

- Nit: Line 752: the symbols used to represent action space in this line are overloaded in the paper. Can we try to use something else here?

- Can we have any specific ablations or experiments to justify or get some research direction that the presented framework is not robust to domain shifts?

- Orthogonal direction: Can this hierarchical framework be applied to more structured observation spaces like unban autonomous driving environments and how good or bad can it be considering that this frame tries to alleviate the exploration problem in policy gradient methods and covariate shifts in imitation learning and other off-policy methods?

---

> ### Author Response · Authors · 2024-11-23
>
> Thanks for your valuable feedback and suggestions. We appreciate the effort and time invested in evaluating our submission. We will address the comments below.
>
> > The pipeline is not able to adapt to different domains, hence generalizability is a problem for now.
>
> > The work seems to be not ready for real-world deployment and is best suited for simulation based research.
>
> > Can we have any specific ablations or experiments to justify or get some research direction that the presented framework is not robust to domain shifts?
>
>
> We did not mean to imply that we had evidence to show we cannot handle domain shifts. We were only observing that we have not yet demonstrated this ability by deploying on a real vehicle. We are planning to deploy it in the future. We have also revised the conclusion section of the paper to clarify this.
>
> > The authors have presented why other IL methods, on-policy methods and off-policy methods might not perform well for off-road terrain considering the diverse scenarios; but it would be good to have some contrastive quantitative metrics to present how TADPO is trying to resolve those challenges, for example handling diverse scenes.
>
>
> In Tables 1 and 2 (as shown in the updated paper and Summary of Changes), we provided comparisons on how TADPO handles various scenarios including extreme slope, obstacles, and hybrid terrain. In this instance, hybrid terrain refers to driving over diverse terrains with different combinations of obstacles, slopes. If there are more metrics that can help illustrate the effectiveness of TADPO, we are happy to provide further data.
>
> > Orthogonal direction: Can this hierarchical framework be applied to more structured observation spaces like unban autonomous driving environments and how good or bad can it be considering that this frame tries to alleviate the exploration problem in policy gradient methods and covariate shifts in imitation learning and other off-policy methods?
>
>
> Though we have not explored how TADPO would perform in environments with more structured observation spaces and even in a multi-agent scenario, we think this method could work well because it bridges the gap between incorporating priors from expert policies and allowing sufficient self-exploration of the student policy to prevent the compounding errors issues one might encounter at test time. Additionally, because the dense planner, the teacher policy, and the student policy can all have different observation spaces, one could leverage privileged information to enable superior performance in both the planner and the teacher policy, allowing the student to learn and explore much more effectively.
>
> Furthermore, we have addressed the style and format concerns raised in your comments. We hope our response clarifies the concerns raised and demonstrates the contributions of our paper. Thank you for your time and consideration.

---

> > ### Comment · Reviewer_BkEv · 2024-11-27
> >
> > Thanking the authors for addressing the questions.
> >
> > I went through the comments and discussions from other reviewers as well and it seems that the authors have tried to carefully address their concerns as well.
> > The authors were able to explain their views regarding some of the concerns related to real-world deployment, generalizability and why MPPI is useful in the proposed work.
> > The authors have presented some more comparisons with other baselines and provided more ablations.
> >
> > It would be nice if the addressed comments can get mentioned in the final version of the paper.
> >
> > I would like to maintain my score considering the limitations pointed out by some other reviewers.
> > Best wishes! :)

---

### Author Response · Authors · 2024-11-23
**Summary of Changes**

Thanks to all reviewers for their time in reviewing this paper and their constructive comments. Based on the comments, we have made the following changes to the paper (highlighted in blue):
- Added new model-based baselines (CEM and RL+MPPI).
-  Included more detailed discussions about the results and added relevant sections in the appendix.
-  Incorporated several related works suggested by the reviewers.
- Revised Tables 1 and 2 to clarify the planners and controllers used in each method.
- Improved clarity by updating the language and equations in multiple sections to align with the aforementioned changes.

We have also attached the updated results tables below for convenience.

### Table 1 (RL and Imitation Learning Baselines)
| Controller        | Planner          | Extreme Slopes | Extreme Slopes | Extreme Slopes | Obstacles | Obstacles | Obstacles | Hybrid   | Hybrid   | Hybrid   |
| ----------------- | ---------------- | -------------- | -------------- | -------------- | --------- | --------- | --------- | -------- | -------- | -------- |
|                   |                  | `sr`           | `cp`           | `ms`           | `sr`      | `cp`      | `ms`      | `sr`     | `cp`     | `ms`     |
| **Teacher**       | **MPPI-d**       | 0.88           | 0.94           | 5.83           | 1.00      | 1.00      | 5.91      | 0.94     | 0.96     | 5.69     |
| | | | | | | | | | | |
| **DAgger**        | **MPPI-s**       | 0.00           | 0.58           | 1.96           | 0.00      | 0.83      | 1.62      | 0.00     | 0.79     | 1.68     |
| **Vanilla PPO**   | **MPPI-s**       | 0.00           | 0.14           | 0.38           | 0.00      | 0.25      | 0.49      | 0.00     | 0.37     | 0.40     |
| **PPO+BC**        | **MPPI-s**       | 0.00           | 0.25           | 0.94           | 0.00      | 0.40      | 0.78      | 0.00     | 0.32     | 0.84     |
| **Vanilla SAC**   | **MPPI-s**       | 0.00           | 0.01           | 1.71           | 0.00      | 0.16      | 1.64      | 0.00     | 0.24     | 1.61     |
| **SAC+Teacher**   | **MPPI-s**       | 0.00           | 0.50           | 1.21           | 0.00      | 0.29      | 1.24      | 0.00     | 0.58     | 1.24     |
| **IQL**           | **MPPI-s**       | 0.25           | 0.49           | 4.85           | 0.13      | 0.71      | 5.01      | 0.07     | 0.76     | 5.03     |
| **TADPO\***   (†) | **MPPI-s\*** (†) | **0.75**       | **0.87**       | **4.99**       | **0.85**  | **0.96**  | **5.26**  | **0.67** | **0.88** | **5.30** |

### Table 2 (Model-based Baselines)
| Planner          | Controller      | Extreme Slopes | Extreme Slopes | Extreme Slopes | Obstacles | Obstacles | Obstacles |  Hybrid  |  Hybrid  |  Hybrid  |   Time    |
| ---------------- | --------------- | :------------: | :------------: | :------------: | :-------: | :-------: | :-------: | :------: | :------: | :------: | :-------: |
|                  |                 |      `sr`      |      `cp`      |      `ms`      |   `sr`    |   `cp`    |   `ms`    |   `sr`   |   `cp`   |   `ms`   |   `ti`    |
| **CEM-d**  | **PID**         |      0.88      |      0.96      |      5.51      |   1.00    |   1.00    |   5.16    |   0.87   |   0.94   |   5.13   |   3.47    |
| **MPPI-d** | **PID**         |      0.88      |      0.96      |      5.39      |   1.00    |   1.00    |   5.87    |   0.87   |   0.94   |   5.43   |   2.02    |
| **RL+MPPI-d**    | **PID**         |      0.88      |      0.96      |      5.26      |   1.00    |   1.00    |   5.88    |   0.87   |   0.94   |   5.40   |   1.77    |
| **MPPI-d**       | **Teacher**     |      0.88      |      0.94      |      5.83      |   1.00    |   1.00    |   5.91    |   0.94   |   0.96   |   5.69   |   2.02    |
| | | | | | | | | | | | |
| **CEM-d\***      | **PID**         |      0.38      |      0.49      |    **5.52**    |   0.25    |   0.38    |   5.16    |   0.27   |   0.43   |   5.13   |   0.13    |
| **MPPI-d\***     | **PID**         |      0.38      |      0.57      |      5.43      |   0.25    |   0.48    | **5.48**  |   0.27   |   0.46   |   5.54   |   0.12    |
| **RL+MPPI-d\***  | **PID**         |      0.38      |      0.61      |      5.32      |   0.25    |   0.50    |   5.46    |   0.27   |   0.52   | **5.63** |   0.12    |
| **MPPI-s\*** (†) | **TADPO\*** (†) |    **0.75**    |    **0.87**    |      4.99      | **0.85**  | **0.96**  |   5.26    | **0.67** | **0.88** |   5.30   | **0.002** |

Our method (†) compared with baselines, where sr denotes success rate, cp denotes average completion percentage, and ms denotes mean speed. MPPI-d refers to the local planner which outputs dense waypoints. MPPI-s refers to the global planner which outputs sparse waypoints. “Extreme Slopes” and “Obstacles” represent the challenging trajectories within the test set, while
“Hybrid” refers to a combination of simpler and difficult trajectories. More information regarding the metrics is in A.7.

---

> ### Comment · Reviewer_BkEv · 2024-11-27
>
> Thank you for providing the updated table. A small suggestion would be to re-arrange the order of 'Controller' and 'Planner' columns in both the tables so that the ordering is consistent to avoid any confusion.

---

### Meta-Review · Area_Chair_uQD1 · 2024-12-19

**Metareview:**

This paper proposes hierarchical controller for off-road driving that incorporates an MPPI mid-level planner with an RL-based low-level controller. A teacher-student framework is proposed. Experiments comparing the algorithms performance on the BeamNG simulator to a few other control algorithms portray the proposed approach favorably.

In terms of strengths, reviewers commented on the novelty of the idea of distilling the teacher into the student via PPO, that the comparison was elaborate, the algorithm is designed to be executed in real-time, that the integration of MMPI with RL in a teacher-student framework is innovative and well-motivated, and that the experiments demonstrate the effectiveness.

In terms of weaknesses, reviewers commented that the pipeline is not able to adapt to different domains, the work is not ready for real-world deployment, that the paper lacked some quantitative metrics to illustrated how the approach resolved challenges that other approaches encountered (e.g. handling diverse scenes), concerns about the correctness of the MMPI description, there are missing descriptions of and missing comparisons to related work, that the experiments do not validate the modified objective over a simple BC loss, issues with description and notation of the algorithm, unclear ability to generalize to real-wold conditions, a reliance on dense teacher guidance, and exploration limitations, incomprehensive selection of baseline methods, and missing complexity analysis to validate one of the claims of the paper.

In response to the author responses, there was unfortunately very limited engagement from the reviewers. The authors responded to each of the stated weaknesses.

After considering the reviews, responses, and my own read of the paper, I believe this paper is a somewhat borderline case. In my opinion, the most salient factor here is the extent to which the stated claims are both novel and evidenced. Unfortunately, the main issue here is the ambiguous novelty of the stated claim. This issue was brought up by reviewer hN5d, to which the authors responded that:
> (i) we created a novel method to train modified PPO (i.e. TADPO) RL policies using teacher demonstrations; and (ii) we combined TADPO with a sparse MPPI planner to achieve real-time off-road driving. Combining these algorithmic contributions, we demonstrate that RL algorithms can be used to effectively learn planning and work synergistically with model-based planning techniques.

The first novelty claim here is of ambiguous importance, as it mentions only that the method is novel. The second novelty claim is that the control is real-time, which, as its a secondary claim and not a substantial contribution, is not well-evidenced just from simulation results (no hardware experiments were conducted). Finally, the summary that "we demonstrated that RL algorithms can be used to effectively learn planning and work synergistically with model-based planning techniques" -- while this may be true, in its current form, this contribution is not novel, as this is essentially what is studied in model-based reinforcement learning, a field with fairly rich literature.

Given the disagreement among reviewers on the evaluation and fairly ambiguous claims of novelty, I do not consider this paper to be acceptable in its current form.

**Additional Comments On Reviewer Discussion:**

See above for points raised by the reviewers. The authors responded by generally clarifying things that were incorrect and included additional experiments. Unfortunately, the reviewers with negative opinions did not engaged with the author responses, leaving us to evaluate the discussion ourselves.

---

### Decision · Program_Chairs · 2025-01-22

Reject